# Antiangiogenic Effects of Coumarins against Cancer: From Chemistry to Medicine

**DOI:** 10.3390/molecules24234278

**Published:** 2019-11-24

**Authors:** Mohammad Bagher Majnooni, Sajad Fakhri, Antonella Smeriglio, Domenico Trombetta, Courtney R. Croley, Piyali Bhattacharyya, Eduardo Sobarzo-Sánchez, Mohammad Hosein Farzaei, Anupam Bishayee

**Affiliations:** 1Student Research Committee, Kermanshah University of Medical Sciences, Kermanshah 6714415153, Iran; mb.majnooni64@yahoo.com; 2Pharmaceutical Sciences Research Center, Health Institute, Kermanshah University of Medical Sciences, Kermanshah 6734667149, Iran; pharmacy.sajad@yahoo.com; 3Department of Chemical, Biological, Pharmaceutical and Environmental Sciences, University of Messina, Viale Palatucci, 98168 Messina, Italy; asmeriglio@unime.it (A.S.); domenico.trombetta@unime.it (D.T.); 4Lake Erie College of Osteopathic Medicine, Bradenton, FL 34211, USA; CCroley48578@med.lecom.edu; 5Escuela de Ciencias de la Salud, Universidad Ana G. Méndez, Recinto de Gurabo, Gurabo, PR 00778, USA; pbhattacharyya@suagm.edu; 6Laboratory of Pharmaceutical Chemistry, Department of Organic Chemistry, Faculty of Pharmacy, University of Santiago de Compostela, 15782 Santiago de Compostela, Spain; e.sobarzo@usc.es or; 7Instituto de Investigación e Innovación en Salud, Facultad de Ciencias de la Salud, Universidad Central de Chile, Santiago 8330507, Chile

**Keywords:** coumarins, antiangiogenic, cancer, natural agents, chemistry, medicine

## Abstract

Angiogenesis, the process of formation and recruitment of new blood vessels from pre-existing vessels, plays an important role in the development of cancer. Therefore, the use of antiangiogenic agents is one of the most critical strategies for the treatment of cancer. In addition, the complexity of cancer pathogenicity raises the need for multi-targeting agents. Coumarins are multi-targeting natural agents belonging to the class of benzopyrones. Coumarins have several biological and pharmacological effects, including antimicrobial, antioxidant, anti-inflammation, anticoagulant, anxiolytic, analgesic, and anticancer properties. Several reports have shown that the anticancer effect of coumarins and their derivatives are mediated through targeting angiogenesis by modulating the functions of vascular endothelial growth factor as well as vascular endothelial growth factor receptor 2, which are involved in cancer pathogenesis. In the present review, we focus on the antiangiogenic effects of coumarins and related structure-activity relationships with particular emphasis on cancer.

## 1. Introduction

Angiogenesis (also known as neovascularization), the growth of blood vessels from the existing vasculature, has been shown to play a critical role in the development of various diseases, including rheumatoid arthritis, diabetic retinopathy, asthma, endometriosis, psoriasis, obesity, and cancer [1,2,3]. Inflammation, tissue ischemia, and hypoxia which cause the release of the angiogenesis factors, such as vascular endothelial growth factor (VEGF), cytokines, cell adhesion molecules, and nitric oxide (NO), are among the most important triggers of angiogenesis [4]. In 1971, Folkman reported that tumor metastasis occurs as a consequence of angiogenesis [5]. This was the starting point for the design and use of bevacizumab, thalidomide, sunitinib, and axitinib as antiangiogenic drugs in the treatment of a variety of cancers [6,7,8]. Considering the crucial role of angiogenesis in the progression of cancer, investigating novel and potential antiangiogenic compounds is of great importance to combat cancer. Several naturally occurring compounds, including vinblastine, vincristine, paclitaxel, were reported as antiangiogenic and anticancer agents. Besides, other natural compounds with antiangiogenic activities, such as resveratrol, artemisinin, boswellic acid, and cannabidiol, have shown enormous potential for cancer prevention and therapy [9,10,11,12,13]. For instance, endocannabinoid 2-arachidonoyl-glycerol showed a promising anticancer effect in several cell lines [14]. Overall, cancer remains a clinical challenge, despite advancements in its treatment. This raises the need to investigate novel multi-target agents to attenuate multiple signaling pathways involved in tumor progression.

Growing evidence has introduced coumarins as potential multi-targeting agents with various pharmacological effects and medicinal uses [15]. Coumarins, with their 2H-1-benzopyran-2-one structure, are natural compounds that exist in various plant families, including Apiaceae, Asteraceae, Fabaceae, Rutaceae, Moraceae, Oleaceae, and Thymelaeaceae [16]. Apiaceae is the greatest family of plants containing coumarin compounds [16]. Also, due to the antioxidant [17], anti-inflammatory [18], anxiolytic [19], analgesic [20], neuroprotective [21], cardioprotective [22], antidiabetic [23], and anticancer [24] activities of coumarins [25], researchers have studied the synthesis of various coumarin derivatives, in addition to their purification from natural sources [26]. Both synthetic and natural coumarins have shown noticeable anticancer effects in vitro and in vivo through various mechanisms [27], including the inhibition of angiogenesis [28,29]. From a mechanistic point of view, some coumarins have shown promising antiangiogenic effects through the interaction with and repression of signaling mediators involved in angiogenesis [30,31].

In this review, we focus on the cellular signaling pathways of angiogenesis and recent pharmacological antiangiogenic agents, emphasizing natural and synthetic coumarins with antiangiogenic effects as well as their pharmacological mechanisms and structure-activity relationship in cancer.

## 2. Angiogenesis: Biology and Cellular Signaling

Angiogenesis could be controlled by achieving a balance among activating cytokines and growth factors on one hand and inhibiting agents on the other, which stimulate or inhibit endothelial cells (ECs), respectively. Proangiogenic agents include growth factors, namely, VEGF, fibroblast growth factors, epidermal growth factor (EGF), transforming growth factor-β (TGF-β), platelet-derived growth factor (PDGF), placental growth factor (PGF), hepatocyte growth factor/scatter factor (HGF/SF), and cytokines, such as tumor necrosis factor-α (TNF-α), colony-stimulating factor-1 (CSF-1), and interleukin-8 (IL-8) [32,33,34,35]. ECs, fibroblasts, platelets, smooth muscle cells, inflammatory cells, and cancer cells are involved in producing angiogenic growth factors and cytokines [4].

VEGF and FGF have been considered as promising antiangiogenic targets [35]. VEGF mainly acts through tyrosine kinase VEGF receptor 2 (VEGFR2) [36]. Furthermore, bioactive lipids, such as prostaglandin E_2_ (PGE_2_), and sphingosine-1-phosphate (S1P), matrix degenerating enzymes, namely, matrix metalloproteinases (MMP) and heparinases, small mediators (e.g., NO, peroxynitrite, serotonin, and histamine), angiopoietins (Ang), and erythropoietin are among other activators of angiogenesis [35]. In order to attract other angiogenesis-stimulating factors, cancer cells induce a situation of hypoxia by increasing the demand for nutrients and oxygen. In the hypoxic condition, hypoxia-inducible factor-1α (HIF-1α), together with released anti-apoptotic factors, growth factors, and cytokines, provokes angiogenesis [4,37].

On the other hand, angiogenesis could be suppressed by inhibiting proteins, which are classified into either direct or indirect angiogenesis inhibitors. The first class of inhibitors directly suppress ECs in the growing vasculature, while the second class indirectly suppress either tumor cells or other tumor-associated stromal cells [35,38]. This direct inhibitory effect could also be mediated by integrin receptors through several intracellular signaling pathways [39]. Angiostatin, endostatin, arrestin, canstatin, and tumstatin are released by the proteolysis of distinct endothelial cell-matrix molecules and prevent vascular ECs from proliferating and migrating in response to angiogenesis inducers [40]. Interferons, retinoic acid, IL-1, IL-12, tissue inhibitor of metalloproteinases, and multimerin 2 are other angiogenesis inhibitors [41,42].

As previously mentioned, angiogenesis is controlled by a balance between activators and inhibitors of angiogenesis. Hypoxia, as a critical determinant, causes an imbalance between activators and inhibitors by inducing the upregulation of HIF-1α, which elevates the expression of pro-angiogenesis agents as well as suppresses the expression of angiogenesis inhibitors [43]. Therefore, all the mediators in these pathways could be therapeutic targets to inhibit angiogenesis.

## 3. Recent Advancement in Pharmacological Antiangiogenic Agents

Several angiogenesis inhibitors have been found since Folkman first presented the concept of introducing angiogenesis inhibitors as anticancer drugs [5]. RNA interference (RNAi) therapy, chimeric antigen receptor T cell therapy, gene therapy, and pharmacological agents are auspicious antiangiogenic interventions [44]. According to the United States Food and Drug Administration (FDA), approved antiangiogenic agents are classified into two major groups, namely, monoclonal antibodies (mAbs) and small molecules [45].

VEGF receptors (VEGFRs) and related downstream signaling pathways are crucial targets of mAbs. Small molecules also target receptors, including PDGFR, VEGFR, Fms-like tyrosine kinase 3, and c-Kit receptor, and signaling proteins such as Raf, mitogen-activated protein kinase (MAPK), mammalian target of rapamycin (mTOR), and phosphoinositide 3-kinases (PI3K). Besides, the antiangiogenic/anticancer effects of FDA-approved herbal drugs, including vinca [46], taxan [47], camptothecins [48], podophyllotoxins [49], and homoharringtonine [50], related receptors, and downstream signaling pathways have now been confirmed [13].

VEGF (A–D), PDGF (A–D), HGF [51,52], and FGF [53,54] bind to VEGFR (1 and 2), PDGFR (*α* and β), MET [55,56], and FGFR (1-4) tyrosine kinase receptors, respectively, and activate downstream signaling pathways, thereby regulating cell growth, differentiation, and angiogenesis [54,57,58,59]. Their overactivation is attributed to several mutations promoting tumor vascularization in different types of cancers [60,61], while their inhibitors exert antitumor effects [62]. Bevacizumab, aflibercept, and ramucirumab have been developed as antiangiogenic agents to target the VEGF/VEGFR signaling pathway [63].

Angiopoietins (Ang1–4) bind to the Tie2 receptor. While Ang1 helps the vessels stabilize, Ang2 is secreted by ECs in response to proangiogenic factors, including hypoxia, cytokines, and inflammation [64]. Ang/Tie2-targeted therapy is challenging, since it could be either antitumor or protumor, depending on the context [65].

The rearranged during transfection (RET) protein binds receptor tyrosine kinases (RTKs) associated with normal development, maintenance, and maturation of cells and tissues [66]. However, its mutation is related to the growth and progression of tumors [66,67]. Therefore, RET inhibition could be of great importance in combating cancer.

Multi-targeting antiangiogenic drugs are shown in Figure 1. These drugs exert anticancer effects through simultaneously modulating several signaling pathways involved in angiogenesis.

## 4. Coumarins

### 4.1. Chemical Structure and Sources

Coumarin (C_9_H_6_O_2_, 2H-1-benzopyran-2-one, 146.145 g/mol) and its derivatives (Figure 2) are a large class of natural compounds that are widely distributed in the plant kingdom and are biosynthesized from ortho-hydroxy-cinnamic acid in the shikimic acid pathways [68]. In terms of chemical structure, coumarins are subdivided into four main groups: (a) simple coumarins, such as heparin and scopoletin; (b) furanocoumarins (linear and angular), such as bergapten and imperatorin; (c) pyranocoumarins, such as grandivittin and agasyllin; (d) dicoumarins and pyrone-substituted coumarins, such as phenylcoumarins (Figure 2) [69,70,71].

Coumarins are isolated and purified from fruits, leaves, stems, roots, and flowers of more than 40 plant families. The Apiaceae represents a family of plants with the highest number of species producing coumarins, including *Anethum graveolens*, *Angelica dahurica*, *Apium graveolens*, *Petroselinum crispum* and *Heracleum mantegazzianum*. Other plant families producing coumarins are Rutaceae (*Citrus aurantium*, *Citrus sinensis*, and *Melicope glabra*), Asteraceae (*Matricaria recutita* and *Achillea millefolium*), Fabaceae (*Melilotus officinalis* and *Glycyrriza glabra*), and Moraceae (*Ficus carica*) [16,72,73,74,75,76].

### 4.2. Biological and Pharmacological Effects of Coumarins

Coumarins have several biological and pharmacological effects. For example, coumarins isolated from the bark of *M. glabra* showed antioxidant properties [73]. In this line, antimicrobial effects of coumarins from the fruits of *H. mantegazzianum* Sommier & Levier as well as *Peucedanum luxurians* Tamamsch were reported [76,77]. Antiviral effects of coumarins isolated from *Prangos ferulacea* L. have been shown by Shokoohinia et al. [78]. In addition, anxiolytic effects of coumarin derivatives, purified from the root of *Biebersteinia multifida* DC, have been demonstrated [79]. Other coumarins, such as umbelliferone and pimpinellin, were isolated from the root of *Zosima absinthifolia* and these compounds showed anti-Alzheimer effects [80]. Kontogiorgis and co-workers [81] designed and synthesized coumarin derivatives based on azomethine, with anti-inflammatory activities. Synthesized coumarins based on 3,4-dihydro-2H-benzothiazines showed analgesic effects in formalin- and acetic acid-induced writhing tests [82]. Additionally, various coumarins have shown antiulcerogenic [83], spasmolytic [84], anticoagulant [85], vasorelaxant [86], cytotoxic, and anticancer activities [87].

On the other hand, hepatotoxicity, nausea, and diarrhea were reported as the side effects of coumarin derivatives [88,89].

### 4.3. Coumarins as Anticancer Agents

As the second leading cause of death worldwide, cancer is one of the most critical diseases that threaten public health and imposes a high cost on countries’ health systems each year. Because of the resistance of cancer cells to conventional drugs used in chemotherapy as well as the side effects of these drugs, it is necessary to find new anticancer agents. Undoubtedly, medicinal plants are one of the richest sources of biologically active compounds and potential novel anticancer drugs. Coumarins are natural compounds with low to moderate side effects, which have been introduced by researchers as promising anticancer compounds [29,90,91,92]. Several coumarins also inhibit cytochrome P450, thereby affecting the blood concentration of various anticancer drugs. In this line, bergamottin inhibits cytochrome P450 and reduces the effects of various carcinogenic agents [93].

The anticancer and cytotoxic activities of synthetic and natural coumarins with different functional groups on their basic structure (Figure 2) have been reported by several investigators. These studies showed the anticancer activity of coumarins against breast cancer [89], colon cancer [94], lung cancer [24], ovarian cancer [95], hepatocellular carcinoma [96], bladder carcinoma [97], leukemia [98], and other types of cancer in vitro and in vivo, via different mechanisms, including free-radical scavenging, antioxidant activity [99], induction of cell cycle arrest [100], interaction with various signaling pathways with important role in cell differentiation and proliferation [101], telomerase and carbonic anhydrase inhibition [102,103], and antiangiogenic activity [104]. For example, Taniguchi and co-workers [105] isolated eight coumarins from the leaves of *Rhizophora mucronata* and reported the anticancer effects of methoxyinophyllum P, calocoumarin B, and calophyllolide against HeLa cells (cervical cancer), with IC_50_ equal to 3.8, 29.9, and 36.4 μM, respectively, and HL-60 cells (promyelocytic leukemia cells) with IC_50_ of 12.9, 2.6, and 2.2 μM, respectively. Also, three hemiterpene ether coumarins, isolated from *Artemisia armeniaca* Lam, showed cytotoxic effects in HL-60 and K562 cells (chronic myelogenous leukemia cells). In their study, armenin showed the highest cytotoxic effect through cycle arrest, with IC_50_ equal to 22.5 and 71.1 µM for K562 and HL-60 cells, respectively [106].

Among coumarins, clausarin, nordentatin, dentatin, and xanthoxyletin, isolated from dichloromethane extraction of root bark of *Clausena harmandiana*, showed high cytotoxic activities [107]. In this study, clausarin (Figure 2) showed the highest cytotoxic activity, which was superior to that of cisplatin used as a positive control, against hepatocellular carcinoma (HepG2, IC_50_ = 17.6 ± 2.1 μM), colorectal carcinoma (HCT116, IC_50_ = 44.9 ± 1.4 μM), and lung adenocarcinoma (SK-LU-1, IC_50_ = 6.9 ± 1.6 μM) cell lines. Clausarin also showed the highest antioxidant activity in the 2,2-diphenyl-1-picryl-hydrazyl-hydrate scavenging assay. From a mechanistic point of view, apoptosis induction was reported as an anticancer mechanism of coumarins [107].

On the other hand, several studies have been carried out on the synthesis of coumarins to produce coumarin derivatives with improved anticancer effects. Among them, synthetic scopoletin derivatives have shown promising antitumor activities. In their study, out of 20 synthesized derivatives of scopoletin, 5 compounds showed the greatest effects (IC_50_ < 2 µM) in MCF-7 and MDA-MB 231 cells (human breast adenocarcinoma cell line) as well as in HT29 cells (human colorectal adenocarcinoma cell line). The relationship between the increase in Log P value and the increase in cytotoxic activity was established in this investigation. Cell cycle arrest was also suggested as the anticancer mechanism of these compounds [108].

Besides, Zang et al. [105] synthesized novel anticancer analogs of geiparvarin using a bioisosteric transformation method. In their study, it was also shown that adding electron-withdrawing substituents to the benzene rings, such as 7-((1-(4-fluorobenzyl)-1*H*-1,2,3-triazol-4-yl)methoxy)-4Hchromen-4-one (Figure 2), increased their cytotoxic effects in a human hepatoma cell line (QGY-7701, IC_50_ = 14.37 ± 9.93) and a colon carcinoma cell line (SW480, IC_50_ = 11.18 ± 2.16) compared with geiparvarin (IC_50_ = 17.68 ± 0.40 and 20.34 ± 0.75, respectively) [109]. Additionally, Cui and co-workers [110] showed the anticancer effects of three synthesized coumarins derived from triphenylethylene, occurring through the inhibition of angiogenesis.

Considering the above investigations on the anticancer effects of coumarin derivatives, these compounds could be developed as anticancer drugs.

## 5. Coumarin and Angiogenesis Inhibition

Inhibition of angiogenesis is one of the most critical anticancer mechanisms of secondary plant metabolites, including coumarins. Natural and synthetic coumarins with different structures can inhibit the factors involved in angiogenesis, migration, proliferation, and differentiation of endothelial cells in vitro and in vivo. Coumarins act by blocking various molecular signaling pathways, involving growth factors (e.g., VEGFs, TNF-α, and FGF-2), cytokines (e.g., IL-1 and IL-6), angiogenic enzymes (e.g., MMP), endothelial-specific receptor tyrosine kinases (e.g., Tie2), and adhesion molecules (e.g., intercellular adhesion molecule-1) [30,111,112].

### 5.1. Natural Coumarins with Antiangiogenic Effects

The antiangiogenic effects of coumarins from natural sources, especially isolated and purified from plants, are reported in several studies. Scopoletin [113,114], esculetin [104], herniarin [115], decursin, and decursinol [116] with coumarin structure, as well as imperatorin [117] and psoralidine [118] with furanocoumarin structure (Figure 2), are among the natural antiangiogenic coumarins.

Pan and co-workers [119] showed the antiangiogenic effects of scopoletin, isolated from *Erycibe obtusifolia* Benth stem, in vitro and in vivo. In their study, scopoletin markedly reduced the number of blood vessel branch points after a 48-h treatment with the dose of 100 nmol/egg in the chick chorioallantoic membrane (CAM) model. Additionally, the inhibitory effects of scopoletin on migration, proliferation, and tube formation of human umbilical vein endothelial cells (HUVECs) induced by VEGF (10 ng/mL) were observed. These investigators showed that the proliferation of HUVECs was significantly inhibited by 100 µM scopoletin after 72 h, and tube formation and migration of HUVECs were inhibited following treatment with 100 µM scopoletin by 52.4% and 38.1%, respectively [119].

Decursin and decursinol angelate, isolated from *Angelica gigas* root, showed substantial antiangiogenic effects in vitro and in vivo [120]. These natural coumarins significantly decreased the development of blood vessels in transgenic zebrafish embryos at 20 µM concentration as well as in the CAM model at 6 µM/egg. In their study, the inhibition of VEGFR2 (one of the most important receptors of VEGF) and other angiogenesis signaling pathway related to VGEF, such as phosphorylated extracellular signal-regulated kinase (p-ERK) and MAPK as well as phosphorylated-c-Jun N-terminal kinase (JNK) in ECs were observed [116].

The antiangiogenic effect of marmesin (Figure 2), a furanocoumarin isolated from ethanolic extract of the twigs of *Broussonetia kazinoki*, was reported by Kim et al. [121]. They showed that marmesin at 10 µM significantly inhibited the expression and activity of MMP-2 in response to VEGF-A (10 ng/mL) in HUVECs. Besides, marmesin reduced EC proliferation, migration, invasion, tube formation and also induced cell cycle arrest in a concentration-dependent manner. Marmesin at 10 µM also inhibited the development of angiogenesis in the rat aortic ring model. VEGF-A stimulated various critical molecules of angiogenesis signaling pathways, such as focal adhesion kinase (FAK), Src kinase, MEK, ERK, Akt, and p70S6K. These pathways were inhibited by 10 µM marmesin [121]. In another study, osthol, columbianadin and columbianetin acetate, three coumarins isolated from *Angelicae Pubescentis* Radix, showed inhibitory effects on the secretion of monocyte chemoattractant protein-1, a pro-inflammation factor and one of the most important migration-regulating chemokines [122]. On the other hand, conferone, a sesquiterpene coumarin isolated from *Ferula szwitziana,* showed antiangiogenic and cytotoxic effects on a human colorectal adenocarcinoma cell line (HT-29) through reducing proangiogenic factors, including angiopoietin 1 and 2 [123]. Besides, daphnetin (7,8-hydroxy coumarin), another natural coumarin, inhibited the expression of MAPK, VEGFR2, ERK1/2, AKT, FAK, and cSrc, which are involved in angiogenesis [124] (Table 1). Also, inhibiting PI3K/AKT activity, another angiogenesis inducer pathway, is one of the antiangiogenic mechanisms of murrangatin purified from *Micromelum falcatum* [125]. Moreover, reducing and blocking VEGF and MMPs are among the most important antiangiogenic mechanisms of coumarins such as galbanic acid [126], umbelliprenin [127], imperatorin [117], auraptene [128], esculetin [31], osthole [129], and scopolin [130] (Table 1).

### 5.2. Synthetic Coumarins with Antiangiogenic Effects

Angiogenesis is a critical process in the development and progression of cancer, as demonstrated by various pre-clinical and clinical evidence. Among natural-based entities, coumarins present little cytotoxicity, while demonstrating more powerful antiangiogenic effects than conventional cytotoxic drugs [131,132]. Semi-synthesized and synthesized products from natural coumarins, used as lead compounds, led to the discovery of interesting antiangiogenic and non-cytotoxic molecules. Coumarins with interesting antiangiogenic and non-cytotoxic properties almost entirely mimic the behavior of the physiological ligands of the main therapeutic targets [132]. Recently, several sulfonyl derivatives of coumarins have been studied as cytotoxic and antiangiogenic agents against HepG2 hepatocellular carcinoma cells in vitro. All synthesized coumarins showed no cytotoxic effect but exhibited a high antimigration activity through the inhibition of MMP-2. CD105 was over-expressed in all cases and, therefore, was not involved in the antimigration activity [132]. In other cases, no statistically significant difference in gene expression of CD44 was found. A synthesized coumarin, 2-oxo-2*H*-chromene-6-sulfonyl derivative, was found to be the most promising antiangiogenic agent, since it was able to inhibit the migratory activity mediated by MMP-2 and down-regulate CD105; however, it did not show any effect on CD44.

On the other hand, the antiangiogenic capacity of several sulfonyl derivatives of coumarins was evaluated using molecular docking studies. In these studies, it was also observed that the compounds showed better docking scores with respect to the I52 ligand, with the nitro derivatives being the best, due to the ability of the nitro group to better coordinate the Zn^++^ ion within the binding site. However, the in vitro antiangiogenic activity of sulfonyl derivatives of coumarins was not statistically significant. Only the 2-oxo-2H-chromene-6-sulfonyl derivative with *N*-acetylpyrazolone substitution at the 6-position showed a promising antiangiogenic activity, exhibiting better binding interactions with the active site and a docking score comparable to that of the inhibitor I52 (−16.22 vs. 18.18 kJmol-1, respectively) [132].

NF-κB is another protein factor, which plays a pivotal role in gene expression and, therefore, is involved in proliferation, angiogenesis, and metastasis, as well as in drug resistance in cancer. In light of this, the development of angiogenesis inhibitors is of significant importance in the treatment of many cancers. Recently, the effects of 26 new synthetic coumarins were tested against hepatocellular carcinoma cells [133]. The investigators identified (7-carbethoxyamino-2-oxo-2*H*-chromen-4-yl)-methylpyrrolidine-1-carbodithioate as the most promising one, because it was cytotoxic in a time- and concentration-dependent manner and it was able to hinder the binding of NF-κB to DNA, therefore inhibiting the expression of several genes, such as *cyclin D1*, *Bcl-2*, *survivin*, *MMP12*, and c-*Myc*. Furthermore, it was able to reduce cell migration and invasion induced by CXCL12, a cytokine that plays a pivotal role in angiogenesis by recruiting endothelial progenitor cells from the bone marrow [133].

Analysis of data present in the existing literature shows that it is possible to identify, through structure-activity analysis, coumarin derivatives more suitable for a given type of tumor. The coumarin derivatives that possess an N-aryl carboxamide, a phenyl substitution at the C-3 position, and 1, 2, 3-triazolyl, trihydroxystilbene, and amino substitutions at the C-4 position were the most effective in targeting lung cancer [24].

Preliminary in vitro results revealed that some coumarin-tethered isoxazolines exhibited significant antiproliferative effect against a human melanoma cancer cell line (UACC 903). Only one derivative with a 3,4-dimethoxy substitution did not show any cytotoxicity against a normal fibroblast cell line (FF2441) in the same concentration range. These results were corroborated in the Ehrlich ascites carcinoma animal model, highlighting significantly lowered cell viability, body weight, ascites volume as well as a down-regulation of angiogenesis and tumor growth [134].

Histone deacetylase 1 (HDAC1), a key element in the control of cell proliferation and differentiation as well as in angiogenesis, represents an attractive therapeutic target for new inhibitors of angiogenesis. In this regard, the benzamidic derivatives of coumarins were found to be the most promising candidates. Four compounds of *N*-(4-((2-aminophenyl)carbamoyl) benzyl)-2-oxo-2*H*-chromene-3-carboxamide derivatives showed the most promising cytotoxic effect, calculated as IC_50_ in the range of 0.53–57.59 μM, on several cancer cells, including HCT116, A2780, MCF7, PC-3, HL60, and A549, without any effects on a human normal cell line (HUVEC, IC_50_ > 100 μM). Moreover, they showed a strong HDAC1-inhibitory activity (IC_50_ 0.47–0.87 μM) with *N*-(4-((2-aminophenyl) carbamoyl) benzyl)-7-((3,4-dichlorobenzyl)oxy)-2-oxo-2*H*-chromene-3-carboxamide, showing an IC_50_ value similar to that of the reference drug entinostat (0.47 ± 0.02 μM vs. 0.41 ± 0.06 μM) [135].

Another interesting coumarin derivative investigated is (*E*)-2-(4-methoxybenzyloxy)-3-prenyl-4-methoxy-N-hydroxycinamide (BMX), a semisynthetic derivative of osthole and a coumarin found in several plant species, such as *Cnidium monnieri* L, *Angelica archangelica* L, and *Angelica pubescens* Maxim. BMX was found to inhibit VEGF-induced proliferation, migration, and endothelial tube formation in HUVECs. These activities were also corroborated by ex vivo and in vivo studies decreasing VEGF-induced microvessel sprouting from aortic rings and HCT116 colorectal cancer cells. Moreover, BMX inhibited HCT116 cell proliferation and the growth of xenografts of HCT116 cells in vivo [136].

Scopoletin is a well-known natural coumarin with antiangiogenic properties. To develop new and robust angiogenesis inhibitors, several scopoletin derivatives were designed and synthesized. According to the study of Tabana et al. [137], scopoletin inhibited VEGF-A, ERK1, and FGF-2, and is thereby considered a strong antiangiogenic agent. In another study, among several scopoletin analogs, three compounds, including 4-bromo-phenyl and 4-chloro-phenyl scopoletin derivatives and 2-hydroxy-3-(piperidin-1-yl)-propoxy)-6-methoxy-2*H*-chromen-2-one, were able to inhibit VEGF-stimulated proliferation, migration, and tube formation of HUVECs. These results showed a significant decrease in the VEGF-triggered phosphorylated forms of ERK1/2 and Akt, which was corroborated by in vivo observations on chick chorioallantoic membrane [138]. Luo and co-workers also showed that 3-aryl-4-anilino/aryloxy-2*H*-chromen-2-one analogues significantly affected breast cancer through the inhibition of estrogen receptor-α and VEGFR-2 [139]. In another study, coumarin-conjugated benzophenone analogs showed promising antitumor activity against Ehrlich ascites carcinoma and Dalton’s lymphoma ascites cell lines. In this study, a compound with a bromo group in the benzophenone structure markedly showed antiangiogenic effects through the inhibition of VEGF [28]. In a more recent study by Makowska et al. [140], a series of 2-imino-2*H*-chromen-3-yl-1,3,5-triazine compounds were synthesized. Among them, 4-[7-(diethylamino)-2-imino-2*H*-chromen-3-yl]-6-(4-phenylpiperazin-1-yl)-1,3,5-triazin-2-amine showed the greatest cytotoxic effects against several human cancer cell lines, which underscores the promising role of synthetic coumarins in combating cancer. Figure 1 illustrates how coumarins inhibit various angiogenic signaling pathways.

## 6. Conclusions

Considering the crucial role of angiogenesis in cancer development, antiangiogenic agents have significant potential to fight cancer. Thus, investigating novel drugs to attenuate or prevent angiogenesis-associated complications in cancer is of great importance. The several clinical limitations and side effects related to the administration of current antiangiogenic agents for cancer treatment raise the need to find alternative treatments. Natural and synthetic coumarins have shown a variety of pharmacological properties. They have demonstrated prominent anticancer effects by targeting multiple signaling pathways involved in several types of cancer.

Recently, studies have focused on the antiangiogenic effects of coumarins according to their structure-activity relationships. The present review reports the currently available literature data on the signaling and regulatory pathways of angiogenesis, as well as on antiangiogenic and anticancer mechanisms of natural and synthetic coumarins, critically analyzing and highlighting their use as possible therapeutic strategies. These studies are essential to identify novel and effective anticancer agents with fewer side effects than conventional drugs. It is also critical to identify potential synergies that may allow reducing the side effects of cytotoxic medicines and increasing the quality of life of patients. Additional studies should focus on additional in vitro and in vivo experiments followed by well-controlled clinical trials to reveal the exact signaling pathways involved in cancer angiogenesis as well as the precise pharmacological mechanisms of coumarins. In addition, there is a need to investigate and adjust novel antiangiogenic coumarin lead compounds to develop more potent and efficient anticancer drugs with lower toxicity. In addition, an appropriate drug delivery system should be introduced to overcome the existing pharmacokinetic challenges of coumarin administration. Such research will unveil the potential of coumarins in the prevention, attenuation, and treatment of angiogenesis in cancer.

## Figures and Tables

**Figure 1 molecules-24-04278-f001:**
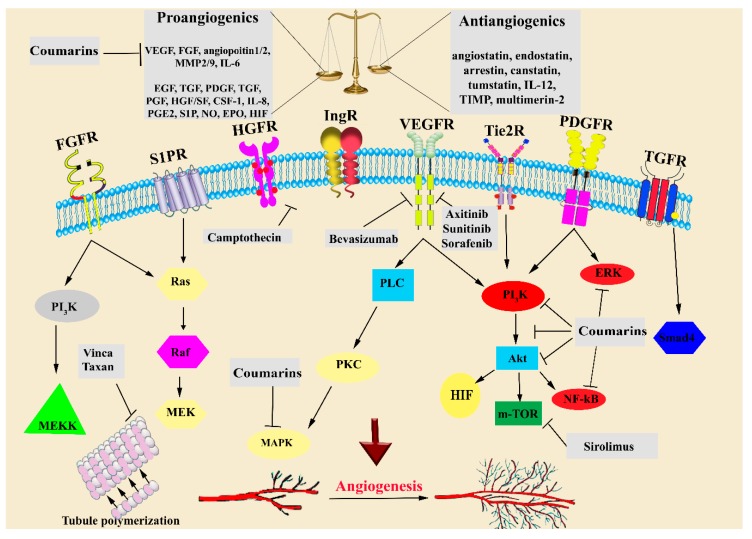
Signaling pathways and therapeutic targets of antiangiogenic and anticancer drugs and agents. VEGF, vascular endothelial growth factor; FGF, fibroblast growth factor; EGF, epidermal growth factor; TGF-β, transforming growth factor-β; PDGF, platelet-derived growth factor; PGF, placental growth factor; HGF/SF, hepatocyte growth factor/scatter factor; TNF-α, tumor necrosis factor-α; CSF-1, colony-stimulating factor-1; IL, interleukin; MMP, matrix metalloproteinase; TIMPs, tissue inhibitors of metalloproteinases; S1PR, sphingosine-1-phosphate receptor; NO, nitric oxide; PI3K:,phosphatidylinositol-3-kinase; PLC, phospholipase C; PKC, protein kinase C; HIF, hypoxia-inducible factor; and m-TOR: mammalian target of rapamycin.

**Figure 2 molecules-24-04278-f002:**
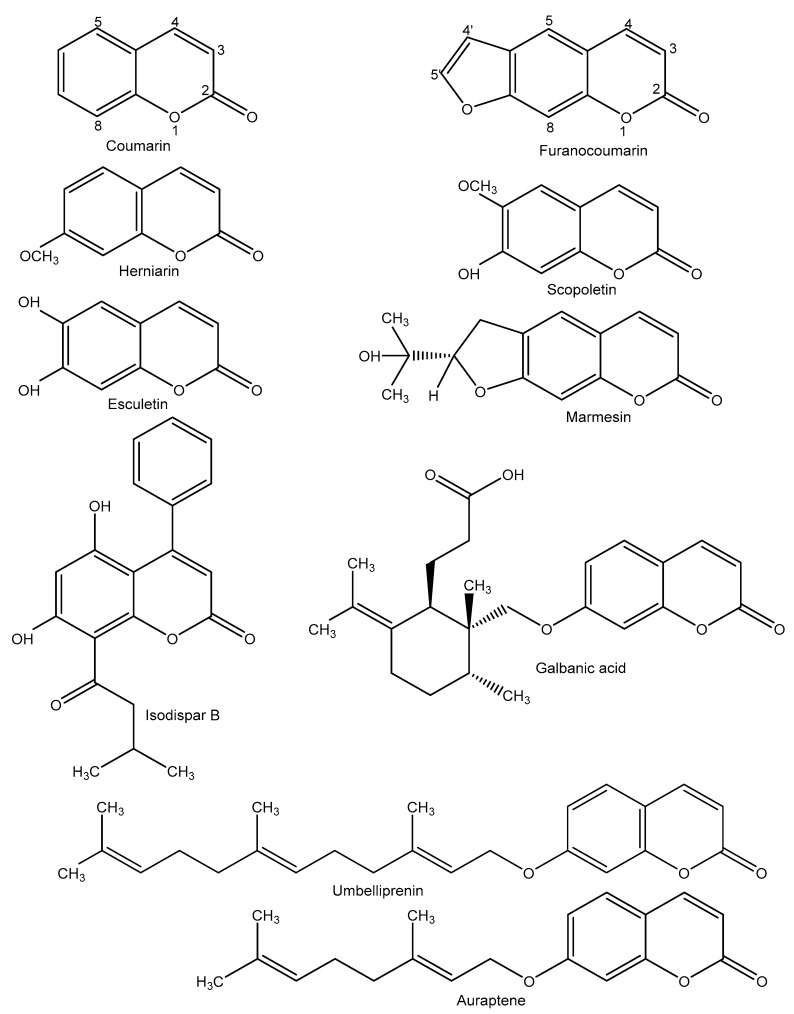
Chemical structures of coumarins with antiangiogenic effects.

**Table 1 molecules-24-04278-t001:** Natural coumarins with antiangiogenic effects and their mechanisms.

Coumarin	Sources	Mechanism of Action	Concentrations	Reference
Galbanic acid	*Ferula assafoetida*	Decreases molecular angiogenesis pathways including VEGF, MAPK, JNK, and AKT	5, 10, 20, 40 µM for in vitro study and 0.1, 1 mg/Kg for Lewis lung cancer (LLC) mouse model, intraperitoneally (i.p.) injected once daily for 18 days	[126]
Umbelliprenin	*Ferula* species	Reduces and blocks angiogenesis marker activity including Ki-67, CD31, VEGF, MMP2, MMP9, and E-cadherin	3, 6.25, 12.5, 25, 50, 100, and 200 µg/mL	[127]
Murrangatin	*Micromelum falcatum*	Inhibition of phosphoinositol 3-kinase (PI3K)/AKT activity as angiogenic inducer	10, 50, 100 µM	[125]
Imperatorin	*Angelica dahurica* and *Angelica archangelica*	Blocks the expression of nuclear factor (NF-κB) target genes, such as MMP-9, VEGF, IL-6 that are induced by TNF-α	50, 100, 150 µM	[117]
Auraptene	*Citrus sinensis*	Inhibition of angiogenesis via suppression of MMP-2,9	12.5, 25, 50, 100 µM	[128]
Esculetin	*Artemisia scoparia*	Inhibition of angiogenesis via decreasing MMP expression and blocking phosphorylation of VEGFR-2, ERK1/2, Akt, and endothelial nitric oxide synthase (eNOS) induced by VEGF (20 ng/mL)	12.5, 25, 50 µg/ml	[31]
Osthol	*Cnidium monnieri*	Reduction of microvessel density (MVD) with blocked expression of VEGF and NF-κB	61, 122, and 244 mg/kg, i.p, once daily for two weeks in a mouse model of hepatocellular carcinoma (HCC)	[129]
Conferone	*Ferula szwitsiana*	Reduction of the production of the pro-angiogenic factors VEGF, Angiopoietin-1, and Angiopoietin -2	20 µM	[123]
Daphnetin	*Changbai daphne*	Inhibition of the expression of proteins involved in angiogenesis induced by VEGF such as MAPK, VEGFR2, ERK1/2, AKT, focal adhesion kinase (FAK), cSrc, and MMP and inhibition of NF-κB induced by TNF-α	37.5, 75, 150 µM	[124]
Scopolin	*Erycibe obtusifolia* Benth	Decreases VEGF, FGF-2 and IL-6 expressions	25, 50, 100 mg/kg, i.p, once daily for 10 days	[130]

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
