# Peer review of "Antiangiogenic Effects of Coumarins against Cancer: From Chemistry to Medicine"

_molecules, 2019, doi:10.3390/molecules24234278_

Round 1

Reviewer 1 Report

The review is very interesting and highlights how the plant world is a source of active ingredients for treating various pathologies, specifically cancer.

I suggest the authors also describe the other properties of the Coumarines. see the works below as an example

Kumar S, Kumari R, Mishra S. Pharmacological properties and their medicinal

uses of Cinnamomum: a review. J Pharm Pharmacol. 2019 Oct 23.

Jantamat P, Weerapreeyakul N, Puthongking P. Cytotoxicity and ApoptosisInduction of Coumarins and Carbazole Alkaloids from Clausena harmandiana.Molecules. 2019 Sep 18;24(18).  

Yang Y, Zhu R, Li J, Yang X, He J, Wang H, Chang Y. Separation and Enrichment of Three Coumarins from Angelicae Pubescentis Radix by Macroporous Resin withPreparative HPLC and Evaluation of Their Anti-Inflammatory Activity. Molecules.2019 Jul 23;24(14).  

Foroozesh M, Sridhar J, Goyal N, Liu J. Coumarins and P450s, Studies Reported to-Date. Molecules. 2019 Apr 24;24(8).

Karakaya S, Koca M, Yılmaz SV, Yıldırım K, Pınar NM, Demirci B, Brestic M,Sytar O. Molecular Docking Studies of Coumarins Isolated from Extracts andEssential Oils of Zosima absinthifolia Link as Potential Inhibitors forAlzheimer's Disease. Molecules. 2019 Feb 17;24(4).  

Makowska A, Sączewski F, Bednarski PJ, Sączewski J, Balewski Ł. HybridMolecules Composed of 2,4-Diamino-1,3,5-triazines and 2-Imino-Coumarins andCoumarins. Synthesis and Cytotoxic Properties. Molecules. 2018 Jul 3;23(7). 

I also suggest that the authors also highlight other drugs that are used for cancer, eg (cannabinoids)

 Pagano E, Borrelli F, Orlando P, Romano B, Monti M, Morbidelli L, Aviello G,Imperatore R, Capasso R, Piscitelli F, Buono L, Di Marzo V, Izzo AA.Pharmacological inhibition of MAGL attenuates experimental colon carcinogenesis. Pharmacol Res. 2017 May;119:227-236.  

Aviello G, Romano B, Borrelli F, Capasso R, Gallo L, Piscitelli F, Di Marzo V, Izzo AA. Chemopreventive effect of the non-psychotropic phytocannabinoid cannabidiol on experimental colon cancer. J Mol Med (Berl). 2012 Aug;90(8):925-34.

Author Response

Reviewer 1

The authors of this manuscript express their sincere thanks to the reviewer for the critical assessment of our work. The authors have acted upon the recommendations of the reviewer which have resulted in a significant enhancement of the quality of this manuscript. All modifications incorporated in the manuscript are highlighted in red color. A “point-by-point” response to the reviewer's comments is outlined below.

Comment 1:

The review is very interesting and highlights how the plant world is a source of active ingredients for treating various pathologies, specifically cancer. I suggest the authors also describe the other properties of the Coumarines. see the works below as an example:

-Kumar S, Kumari R, Mishra S. Pharmacological properties and their medicinal uses of Cinnamomum: a review. J Pharm Pharmacol. 2019 Oct 23.

-Jantamat P, Weerapreeyakul N, Puthongking P. Cytotoxicity and Apoptosis Induction of Coumarins and Carbazole Alkaloids from Clausena harmandiana. Molecules. 2019 Sep 18;24(18). 

-Yang Y, Zhu R, Li J, Yang X, He J, Wang H, Chang Y. Separation and Enrichment of Three Coumarins from Angelicae Pubescentis Radix by Macroporous Resin withPreparative HPLC and Evaluation of Their Anti-Inflammatory Activity. Molecules.2019 Jul 23;24(14). 

-Foroozesh M, Sridhar J, Goyal N, Liu J. Coumarins and P450s, Studies Reported to-Date. Molecules. 2019 Apr 24;24(8).

-Karakaya S, Koca M, Yılmaz SV, Yıldırım K, Pınar NM, Demirci B, Brestic M,Sytar O. Molecular Docking Studies of Coumarins Isolated from Extracts andEssential Oils of Zosima absinthifolia Link as Potential Inhibitors for Alzheimer's Disease. Molecules. 2019 Feb 17;24(4). 

-Makowska A, Sączewski F, Bednarski PJ, Sączewski J, Balewski Ł. Hybrid Molecules Composed of 2,4-Diamino-1,3,5-triazines and 2-Imino-Coumarins and Coumarins. Synthesis and Cytotoxic Properties. Molecules. 2018 Jul 3;23(7).

Response:

We thank respected reviewer for the generous comments about the quality of our manuscript. The articles the reviewers suggested have been described and cited as follows:

Kumar S, Kumari R, Mishra S. Pharmacological properties and their medicinal uses of Cinnamomum: a review. J Pharm Pharmacol. 2019 Oct 23 – cited in page 2, lines 59 and 60 (reference 15).

Jantamat P, Weerapreeyakul N, Puthongking P. Cytotoxicity and ApoptosisInduction of Coumarins and Carbazole Alkaloids from Clausena harmandiana. Molecules. 2019 Sep 18;24(18) – cited in page 7, lines 245-252 (reference 107).

Yang Y, Zhu R, Li J, Yang X, He J, Wang H, Chang Y. Separation and Enrichment of Three Coumarins from Angelicae Pubescentis Radix by Macroporous Resin withPreparative HPLC and Evaluation of Their Anti-Inflammatory Activity. Molecules.2019 Jul 23;24(14) – cited in page 9, lines 311-314 (reference 122).  

Foroozesh M, Sridhar J, Goyal N, Liu J. Coumarins and P450s, Studies Reported to-Date. Molecules. 2019 Apr 24;24(8) – cited in page 7, lines 225-227 (reference 93).

Karakaya S, Koca M, Yılmaz SV, Yıldırım K, Pınar NM, Demirci B, Brestic M,Sytar O. Molecular Docking Studies of Coumarins Isolated from Extracts and Essential Oils of Zosima absinthifolia Link as Potential Inhibitors for Alzheimer's Disease. Molecules. 2019 Feb 17;24(4) – cited in page 7, lines 208 and 209 (reference 80).

Makowska A, Sączewski F, Bednarski PJ, Sączewski J, Balewski Ł. Hybrid Molecules Composed of 2,4-Diamino-1,3,5-triazines and 2-Imino-Coumarins and Coumarins. Synthesis and Cytotoxic Properties. Molecules. 2018 Jul 3;23(7) – cited in page 12, lines 411-415 (reference 140).

Comment 2:

I also suggest that the authors also highlight other drugs that are used for cancer, eg (cannabinoids)

-Pagano E, Borrelli F, Orlando P, Romano B, Monti M, Morbidelli L, Aviello G,Imperatore R, Capasso R, Piscitelli F, Buono L, Di Marzo V, Izzo AA. Pharmacological inhibition of MAGL attenuates experimental colon carcinogenesis. Pharmacol Res. 2017 May;119:227-236. 

-Aviello G, Romano B, Borrelli F, Capasso R, Gallo L, Piscitelli F, Di Marzo V, Izzo AA. Chemopreventive effect of the non-psychotropic phytocannabinoid cannabidiol on experimental colon cancer. J Mol Med (Berl). 2012 Aug;90(8):925-34.

Response:

We greatly appreciate this suggestion. Other drugs with anticancer effects have been now highlighted as follows:

Pagano E, Borrelli F, Orlando P, Romano B, Monti M, Morbidelli L, Aviello G,Imperatore R, Capasso R, Piscitelli F, Buono L, Di Marzo V, Izzo AA. Pharmacological inhibition of MAGL attenuates experimental colon carcinogenesis. Pharmacol Res. 2017 May;119:227-236 – cited in page 2, lines 54-56 (reference 14). 

Aviello G, Romano B, Borrelli F, Capasso R, Gallo L, Piscitelli F, Di Marzo V, Izzo AA. Chemopreventive effect of the non-psychotropic phytocannabinoid cannabidiol on experimental colon cancer. J Mol Med (Berl). 2012 Aug;90(8):925-34 – cited in page 2, line 54 (reference 9).

Additionally,

We have renumbered the references as we have added numerous new references (indicated by red color font). The manuscript has been edited extensively to improve language and style.

Reviewer 2 Report

The manuscript from Majnooni and coworkers reviews the efficacy of coumarins in antitumor therapy, with particular focus on targets involved in angiogenesis. Both natural and synthetic coumarin derivatives have been considered from literature. The manuscript has a good and fluent style and the topic is pertinent with the scopes of the journal. It can be accepted with minor revisions, as detailed.

From a Pubmed search, some articles strictly related with the topic may be discussed and added (see attached file). Figure 1 is blurred. Structures in Figure 2 are stylistically different, please make them uniform in style and dimension. Uniform the page format (lines 225-232 and 294-302)

Author Response

Reviewer 2

The authors of this manuscript express their sincere thanks to the reviewer for the critical assessment of our work. The authors have acted upon the recommendations of the reviewer which have resulted in a significant enhancement of the quality of this manuscript. All modifications incorporated in the manuscript are highlighted in red color. A “point-by-point” response to the reviewer's comments is outlined below.

Comment 1:

The manuscript from Majnooni and coworkers reviews the efficacy of coumarins in antitumor therapy, with particular focus on targets involved in angiogenesis. Both natural and synthetic coumarin derivatives have been considered from literature. The manuscript has a good and fluent style and the topic is pertinent with the scopes of the journal. It can be accepted with minor revisions, as detailed.

Response:

We express our sincere thanks to the respected reviewer for the encouraging comment about the quality of our manuscript. We sincerely appreciate the suggestions and revised our manuscript accordingly.

Comment 2:

From a Pubmed search, some articles strictly related with the topic may be discussed and added (see attached file as below). Figure 1 is blurred. Structures in Figure 2 are stylistically different, please make them uniform in style and dimension. Uniform the page format (lines 225-232 and 294-302).

References from the attached file:

1: Cui N, Lin DD, Shen Y, Shi JG, Wang B, Zhao MZ, Zheng L, Chen H, Shi JH. Triphenylethylene-Coumarin Hybrid TCH-5c Suppresses Tumorigenic Progression in Breast Cancer Mainly Through Its Inhibition of Angiogenesis. Anticancer Agents Med Chem. 2019 Apr 4. doi: 10.2174/1871520619666190404155230. [Epub ahead of print] PubMed PMID: 30947677.

2: Luo G, Li X, Zhang G, Wu C, Tang Z, Liu L, You Q, Xiang H. Novel SERMs based on 3-aryl-4-aryloxy-2H-chromen-2-one skeleton - A possible way to dual ERα/VEGFR-2 ligands for treatment of breast cancer. Eur J Med Chem. 2017 Nov 10;140:252-273. doi: 10.1016/j.ejmech.2017.09.015. Epub 2017 Sep 14. PubMed PMID: 28942113.

3: Kim JH, Kim MS, Lee BH, Kim JK, Ahn EK, Ko HJ, Cho YR, Lee SJ, Bae GU, Kim YK, Oh JS, Seo DW. Marmesin-mediated suppression of VEGF/VEGFR and integrin β1 expression: Its implication in non-small cell lung cancer cell responses and tumor angiogenesis. Oncol Rep. 2017 Jan;37(1):91-97. doi: 10.3892/or.2016.5245. Epub 2016 Nov 15. PubMed PMID: 27878269.

4: Vijay Avin BR, Thirusangu P, Lakshmi Ranganatha V, Firdouse A, Prabhakar BT, Khanum SA. Synthesis and tumor inhibitory activity of novel coumarin analogs targeting angiogenesis and apoptosis. Eur J Med Chem. 2014 Mar 21;75:211-21. doi: 10.1016/j.ejmech.2014.01.050. Epub 2014 Feb 2. PubMed PMID: 24534537.

Response:

We admire the reviewer for these thought-provoking comments. The references suggested by the reviewer have been discussed and cited the manuscript as follows:

Cui N, Lin DD, Shen Y, Shi JG, Wang B, Zhao MZ, Zheng L, Chen H, Shi JH. Triphenylethylene-Coumarin Hybrid TCH-5c Suppresses Tumorigenic Progression in Breast Cancer Mainly Through Its Inhibition of Angiogenesis. Anticancer Agents Med Chem. 2019 Apr 4. doi: 10.2174/1871520619666190404155230. [Epub ahead of print] PubMed PMID: 30947677 – cited in page 8, lines 267 and 268 (reference 110).

Luo G, Li X, Zhang G, Wu C, Tang Z, Liu L, You Q, Xiang H. Novel SERMs based on 3-aryl-4-aryloxy-2H-chromen-2-one skeleton - A possible way to dual ERα/VEGFR-2 ligands for treatment of breast cancer. Eur J Med Chem. 2017 Nov 10;140:252-273. doi: 10.1016/j.ejmech.2017.09.015. Epub 2017 Sep 14. PubMed PMID: 28942113 – cited in page 12, lines 405-407 (reference 139).

Kim JH, Kim MS, Lee BH, Kim JK, Ahn EK, Ko HJ, Cho YR, Lee SJ, Bae GU, Kim YK, Oh JS, Seo DW. Marmesin-mediated suppression of VEGF/VEGFR and integrin β1 expression: Its implication in non-small cell lung cancer cell responses and tumor angiogenesis. Oncol Rep. 2017 Jan;37(1):91-97. doi: 10.3892/or.2016.5245. Epub 2016 Nov 15. PubMed PMID: 27878269 – cited in page 8, line 304 to page 9, line 312 (reference 121).

Vijay Avin BR, Thirusangu P, Lakshmi Ranganatha V, Firdouse A, Prabhakar BT, Khanum SA. Synthesis and tumor inhibitory activity of novel coumarin analogs targeting angiogenesis and apoptosis. Eur J Med Chem. 2014 Mar 21;75:211-21. doi: 10.1016/j.ejmech.2014.01.050. Epub 2014 Feb 2. PubMed PMID: 24534537 – cited in page 2, line 69 and page 12, lines 407-411 (reference 28).

Figure 1 has been improved to enhance the quality. All structures in Figure 2 (pages 5 and 6) have been redrawn to ensure uniform style and dimension.

We have corrected the formatting errors for the indicated paragraphs (page 7, lines 245-252 and page 9, lines 315-324).

Additionally,

We have renumbered the references as we have added numerous new references (indicated by red color font). The manuscript has been edited extensively to improve language and style.

Reviewer 3 Report

Reviewed manuscript “Antiangiogenic Effects of Coumarins Implicated in Cancer: From Chemistry to Medicine” could be accepted for the publication in Molcules after revision. I am not sure this review could be very interesting to the readers of this journal, at least it was interesting for me. It looks like chapter 1 in some PhD thesis.

It could be nice if respected authors remove sentences from a handbook. First example, lines 76-82. There is introduction of what is angiogenesis.

Lines 190-198 Please re-draw all schemes. It is not in acceptable shape.

Please re-write part (5.2). It is absolutely impossible to read, because there are a lot of complicated IUPAC names (with mistakes). Please add structures and compound numbers.

Author Response

Reviewer 3

The authors of this manuscript express their sincere thanks to the reviewer for the critical assessment of our work. The authors have acted upon the recommendations of the reviewer which have resulted in a significant enhancement of the quality of this manuscript. All modifications incorporated in the manuscript are highlighted in red color. A “point-by-point” response to the reviewer's comments is outlined below.

Comment 1:

Reviewed manuscript “Antiangiogenic Effects of Coumarins Implicated in Cancer: From Chemistry to Medicine” could be accepted for the publication in Molcules after revision. I am not sure this review could be very interesting to the readers of this journal, at least it was interesting for me. It looks like chapter 1 in some PhD thesis.

Response:

We express our sincere thanks to the reviewer for finding our manuscript interesting.

Comment 2:

It could be nice if respected authors remove sentences from a handbook. First example, lines 76-82. There is introduction of what is angiogenesis.

Response:

We have revised and consolidated the first two introductory paragraphs of section 2 (page 2, lines 79-86) as suggested by the reviewer.

Comment 3:

Lines 190-198 Please re-draw all schemes. It is not in acceptable shape.

Response:

We have re-drawn all structures in Figure 2 (pages 5 and 6) to improve quality.

Comment 4:

Please re-write part (5.2). It is absolutely impossible to read, because there are a lot of (with mistakes). Please add structures and compound numbers.

Response:

We express our sincere thanks for this constructive suggestion. Accordingly, we have re-written various parts of section 5.2.  In order to improve the fluency of the text, the complicated IUPAC names have been replaced with coumarin-based derivative names. The IUPAC names with mistakes have been also corrected. All modifications are highlighted in red color font (page 11, lines 342-345; page 11, lines 352-355; page 11, lines 360 and 361; page 11, lines 381-388; and page 12, lines 400-415).

Additionally,

We have renumbered the references as we have added numerous new references (indicated by red color font). The manuscript has been edited extensively to improve language and style.